# Efficient Audiovisual Speech Processing via MUTUD: Multimodal Training and Unimodal Deployment

## Abstract

Building reliable speech systems often requires combining multiple modalities, like audio and visual cues. While such multimodal solutions frequently lead to improvements in performance and may even be critical in certain cases, they come with several constraints such as increased sensory requirements, computational cost, and modality synchronization, to mention a few. These challenges constrain the direct uses of these multimodal solutions in real-world applications. In this work, we develop approaches where the learning happens with all available modalities but the deployment or inference is done with just one or reduced modalities. To do so, we propose a Multimodal Training and Unimodal Deployment (MUTUD) framework which includes a Temporally Aligned Modality feature Estimation (TAME) module that can estimate information from missing modality using modalities present during inference. This innovative approach facilitates the integration of information across different modalities, enhancing the overall inference process by leveraging the strengths of each modality to compensate for the absence of certain modalities during inference. We apply MUTUD to various audiovisual speech tasks and show that it can reduce the performance gap between the multimodal and corresponding unimodal models to a considerable extent. MUTUD can achieves this while reducing the model size and compute compared to multimodal models, in some cases by almost 80%.

## 1 Introduction

Unimodal (audio-only) approaches to well-known speech problems such as speech enhancement, speaker separation, and automatic speech recognition, have made rapid progress using deep learning. At the same time, multimodal approaches to these tasks are also increasingly gaining significance (Mira et al., 2023; Xu et al., 2020; Ma et al., 2021b; Hong et al., 2022; 2023). While the additional modality may come in different forms such as text, contact microphones, IMUs, etc., visual modality is the most widely used in these speech tasks. This bears similarity to humans as we also innately rely on visuals to perceive sounds and speech (Schwartz et al., 2004). In fact, people with hearing impairments have also been shown to rely on visuals for better perception of speech (Burnham et al., 2013). Given the significance of multimodal perception of speech by humans, it is natural that multimodal learning has shown impressive gains over unimodal systems for various speech tasks. The role of visuals in speech understanding becomes much more critical in acoustically difficult scenarios such as noisy environments or situations where the speech signals on their own are not reliable for the task at hand (Weninger et al., 2015; Tan & Wang, 2019; Wang et al., 2020; Braun et al., 2021).

While multimodal systems can extract supplementary and complementary information from different modalities (Baltrušaitis et al., 2018; Lu, 2023), leading to performance improvements, certain challenges with multimodal models can restrain their uses in real-world systems. These include but are not limited to *(1)* Multimodal models are often computationally much more expensive compared to their unimodal counterparts

and the performance gain might not justify the substantial increase in computational cost. This is especially relevant for real-time and on-device applications (e.g., speech enhancement). In fact, in several cases, this can prohibit the deployment of multimodal systems. *(2)* Multimodal data come at a significantly higher cost. Acquisition of multimodal data requires complex sensory devices working together seamlessly. Alignment, synchronization, and annotation efforts in multimodal data are far more challenging than audio-only data. More importantly, such aligned and synchronized multimodal data is required even during inference, necessitating the availability of all sensory devices and the processing power to align and synchronize the captured signals. This can make multimodal systems impractical in several real-world applications. *(3)* Lastly, it might not be feasible to use multiple modalities for a speech task due to practical constraints such as privacy or difficulties in getting signals for all modalities. For example, while multimodal ASR could improve audio-only ASR in noisy conditions, getting the visual signals during real-world uses might not be possible.

The above discussion highlights benefits of multimodal learning defnitely over unimodal learning, yet there are certain constraints which can make unimodal models preferable over multimodal despite lower performance. Motivated by this, the primary question we ask is *how do we learn from multimodal data while enabling unimodal uses of the model?* In this framework, we still want to learn from the rich information available in multimodal data but unimodal inference removes the constraints around uses of the multimodal system. Note that, unlike works on robustness to missing modality we develop a fundamental approach for *MUltimodal Training and Unimodal Deployment (MUTUD, pronounced "muted")*. In modality robustness, the model behavior remains the same during training and inference, and hence the challenges of multimodal systems outlined before are not rectified. MUTUD is driven by architectural and training novelties, which addresses those challenges. MUTUD framework is built using a novel *Temporally Aligned Modality feature Estimation (TAME)* module. The TAME module is designed to estimate deep representations of modalities which are *absent* during inference using the representations of modalities *present* during inference. TAME achieves this by having codebooks for each modality and linking cross-modal pairs of codebooks in a way that enables modality feature recall using the codebooks and the features of available modalities.

We apply our framework for 3 well-known tasks in the speech processing domain and do multimodal (audiovisual (AV)) training and unimodal inference; speech enhancement, speech recognition, and active speaker detection. Speech enhancement in particular may have tight real-time and low-compute requirements for several applications. In all the tasks, we show that MUTUD achieves unimodal inference with a significantly better performance compared to the counterpart models trained on unimodal data. Moreover, compared to the full multimodal systems, our model has significantly lesser parameters and compute and yet gives competitive performance.

## 2 RELATED WORKS

**Audiovisual speech processing.** Analogous to humans, AV learning for speech-related tasks naturally results in methods that are more robust to noisy scenarios such as acoustic SNR degradation, poor lighting conditions, motion blur, etc. In this paper we focus on three AV speech problems namely, speech enhancement (Gabbay et al., 2017; Afouras et al., 2018a; Gao & Grauman, 2021; Mira et al., 2023; Yang et al., 2022; Owens & Efros, 2018; Hou et al., 2018), speech recognition (Huang & Kingsbury, 2013; Mroueh et al., 2015; Noda et al., 2015; Stewart et al., 2013; Ma et al., 2021b) and speaker detection (Garg et al., 2000; Cutler & Davis, 2000; Chakravarty et al., 2016; Roth et al., 2020). The reader is referred to excellent survey papers for a detailed overview of different methodologies (Michelsanti et al., 2021; Potamianos et al., 2017). As already highlighted, traditional AV approaches suffer from several constraints such as sensor requirements, computational cost and modality synchronization which limit their applicability in real-world applications.

**Resource-constrained learning.** Considerable progress has been made in resource-constrained audio-only speech processing (Kim et al., 2020; Lee et al., 2021; Maayah et al., 2023), even though such multimodal methods are relatively smaller. Typical strategies include lightweight network design (Maayah et al., 2023), quantization and pruning (Tan et al., 2021) and knowledge distillation (Thakker et al., 2022). Gogate

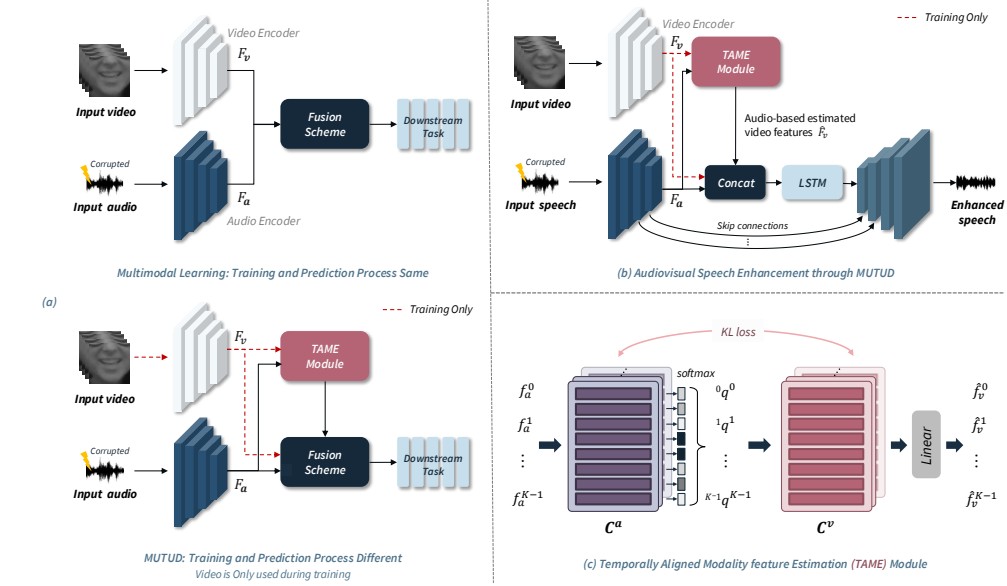

Figure 1: (a) The left panel shows a comparison between conventional audiovisual speech processing and MUTUD. TAME module enables audiovisual learning without doing video processing during prediction. (b) The upper half in the right panel illustrates MUTUD for an AVSE model. After training the video encoder is discarded. (c) The bottom half in the right panel shows the estimation of video representations using TAME. The illustration is for $t = 0$ in Eq 4.

et al. (2020) build a robust language-dependent audiovisual model called CochleaNet for real-time speech enhancement through audiovisual mask estimation. LAVSE (Chuang et al., 2020) proposed a visual data compression technique for speech enhancement. Our focus in this work is very different. We intend to develop efficiency in multimodal learning by allowing resource-heavy modalities to be absent during prediction or when deployed.

**Learning with missing modality.** Multimodal learning for robustness to missing modality is a practical problem that has been explored in some works before. Each work differs in the modality considered to be missing, the phase (training or testing) in which this information is absent, and whether the loss of information is partial or complete (Hegde et al., 2021; Ma et al., 2021a; Woo et al., 2023; Ma et al., 2022; Lee et al., 2023). The methods are often tailor-made for the scenarios in consideration. For brevity, here we limit our discussion to AV speech-related tasks. Some studies rely on a memory architecture to retrieve missing modality via associated bridging mechanism (Kim et al., 2021b; Hong et al., 2021; Kim et al., 2021a). These related works serve as inspiration for MUTUD. Further, AV-HuBERT (Shi et al., 2022) and u-HuBERT (Hsu & Shi, 2022) presented a self-supervised pre-training framework that can leverage both multimodal and unimodal speech with a unified masked cluster prediction objective, achieving zero-shot modality generalization for multiple speech processing tasks. While these works have made significant progress on various speech processing problems, they are very different from us – their focus is on self-supervised training of large models with massive amounts of unlabeled data. The learned models are then fine-tuned for tasks like ASR. These models are not designed for unimodal deployment with compute/memory efficiency in mind. Furthermore, it is difficult to adapt AV-HuBERT/u-HuBERT for tasks like speech enhancement, especially in causal settings.

Unlike these works, we are driven by the challenges of multimodal learning outlined before. We focus explicitly on multimodal learning for unimodal prediction and real-world deployment, which addresses those challenges. Our approach is fairly generic and can be applied to many common multimodal learning methods and tasks.

# 3 MUTUD: Multimodal Training and Unimodal Deployment

We describe our proposed method, which we call Multimodal Training and Unimodal Deployment (MUTUD). Our goal is to design a network that leverages multimodal sensory inputs during training, but only takes in a subset of them during inference. In section 3.1, we first describe MUTUD in its general setting, where an arbitrary number of modalities are considered, followed by a discussion targeted to the audiovisual speech domain. In section 3.2, we introduce our proposed TAME Module, which is the key component to enable unimodal predictions. Finally in section 3.3, we describe the training objectives. The left panel in Figure 1 shows the difference between MUTUD and conventional multimodal learning.

## 3.1 MUTUD Overview

Let $\mathcal{D}$ be a dataset where each sample $X \in \mathcal{D}$ is characterized by $M$ different modalities $X = \{X_{m_i}\}$, $i = 1 : M$. $\mathcal{M} = \{m_1, m_2, \cdots, m_M\}$ is the set of modalities. Conventionally, multimodal learning operates with the assumption that the model always inputs all $M$ modalities, during training as well as for predictions. Let $h^{\mathcal{M}}(X = \{X_{m_i}, m_i \in \mathcal{M}\}; \phi)$ a deep neural network (DNN) based multimodal system (parameterized by $\phi$). In MUTUD, all $M$ modalities of $X$ are available during training, but only a subset, $\mathcal{M}_s \subset \mathcal{M}$, are available during real-world deployment or for inference.

To this end, we design MUTUD with two crucial characteristics in mind. Let $h(; \theta)$ be the MUTUD system. (1) Since $h(; \theta)$ processes only $|\mathcal{M}_s|$ modalities for prediction, we expect it to have fewer parameters and be computationally more efficient in real-world deployment. Ideally, we would like $h(; \theta)$ to have inference size and compute similar to $h^{\mathcal{M}_s}(X = \{X_{m_i}, m_i \in \mathcal{M}_s\}; \psi)$, a model counterpart of $h^{\mathcal{M}}(X; \phi)$ with $\mathcal{M} = \mathcal{M}_s$. (2) On the performance end, $h^{\mathcal{M}}(; \phi)$ should have superior performance compared to $h^{\mathcal{M}_s}(; \psi)$ due to utilization of more modalities in the learning process. We expect $h(; \theta)$ to have superior performance compared to $h^{\mathcal{M}_s}(X; \psi)$ and closer to that of $h^{\mathcal{M}}(X; \phi)$.

In a typical multimodal model, all $X_{m_i}$ are encoded by a network, these representations are then fused through various mechanisms (concatenation, attention, etc. (Kalkhorani et al., 2023; Wei et al., 2020; Ma et al., 2021b; Lee et al., 2020; Praveen et al., 2023)). The fused representations are further processed by more neural layers to solve the task at hand. We operate in a similar setting. To achieve our goal, we develop an efficient and effective mechanism to *associate* and *relate* missing modalities, $(\mathcal{M} - \mathcal{M}_s)$, to those in $\mathcal{M}_s$, such that the representations of $X_{m_i} \in \mathcal{M} - \mathcal{M}_s$ can be recalled using those of $X_{m_i} \in \mathcal{M}_s$. We propose a Temporally Aligned Modality feature Estimation (TAME) module. TAME learns a pair of codebooks ($C^{m_i}$, $C^{m_j}$) for each pair of modality in $\{(m_i, m_j), \forall m_i \in \mathcal{M} - \mathcal{M}_s, \forall m_j \in \mathcal{M}_s\}$. The training objectives link these codebooks in a way that enables estimation of representations for $X_{m_i} \in \mathcal{M} - \mathcal{M}_s$ during inference.

**MUTUD for AudioVisual Speech Processing** We focus on audiovisual speech tasks where MUTUD is designed to use only one of them during deployment. For a succinct and clear description of MUTUD and TAME, we explain it through the task of Audiovisual Speech Enhancement (AVSE) but the method similarly adapts to other tasks. The right panel in Figure 1 outlines the base AVSE model ($h^{\mathcal{M}}(; \phi)$). The speech and video encoders produce $F_a \in \mathbb{R}^{T_a \times D}$ and $F_v \in \mathbb{R}^{T_v \times D}$ representations, respectively. $T_a$ and $T_v$ represent time dimensions and depend on the frame rates of speech and audio. The frame rate of speech is $K$ times of video ($T_a = T_v * K$) and hence $F_v$ is upsampled by a factor of $K$ to match the size along a temporal direction before the concatenation step. The concatenated representations are then decoded by the decoder to produce the enhanced speech. The $h^{\mathcal{M}_s}(; \psi)$ model is the audio-only model, where everything is the same except that there are no visual inputs, and the decoder decodes the encoded audio representations to output enhanced speech. Under MUTUD, our goal is to train with both visual and audio inputs but deploy an audio-only model. Hence, we design and train TAME module to estimate video representations during prediction.

## 3.2 TAME Module

The core of the TAME consists of modality-specific codebooks (MSCs) for audio and video. These are used to associate and relate modalities through their respective representations during training. During inference,

the audio representations are used to retrieve the video representations through these MSCs. The MSCs are designed to capture temporal alignment and synchronized relations between the audio and the video. Since the audio representations frame rate (in $F_a$) is higher by a factor of $K$, we design TAME keeping this temporal relation in consideration. That is the $t^{th}$ video frame feature in $F_v$, $f_v^t$, is associated with $K$ audio features $(f_a^{K \cdot t}, f_a^{K \cdot t+1}, \ldots, f_a^{K \cdot t+K-1})$ in $F_a$. Besides keeping the temporal alignment between audio and video representations intact, this temporal coupling between the audio and video is also necessary for learning to estimate video features using audio.

TAME formulates this through $K$ blocks of codebooks in each MSC, represented as $\boldsymbol{C}^a \in \mathbb{R}^{K \times N \times D}$ and $\boldsymbol{C}^v \in \mathbb{R}^{K \times N \times D}$ for the audio and video respectively, (see Figure 1). $N$ is the number of codes in each set of codebooks in $\boldsymbol{C}^a$ and $\boldsymbol{C}^v$.

All features in consideration ($f_v^t$ for video and $\boldsymbol{f}_a^t = \{f_a^{K \cdot t}, f_a^{K \cdot t+1}, \ldots, f_a^{K \cdot t+k}, \ldots, f_a^{K \cdot t+K-1}\}$ for audio) are first embedded through their respective MSC. This relationship between $f_v^t$ and $k^{th}$ codebook in $\boldsymbol{C}^v$ is established through the vectors ${}^k v^t$,

$$
{}^k v^t = [\; \frac{<\; {}^k c_n^v, f_v^t\; >}{\|{}^k c_n^v\|_2 \; \|f_v^t\|_2} \;], \;\; \text{where} \;\; {}^k c_n^v = \boldsymbol{C}^v[k, n, :], \;\; n = \{0, 1, \ldots, N\} \tag{1}
$$

${}^k v^t$ is computed for all K codebooks ($k \in \{0, K-1\}$) using Eq 1. Similarly, the audio features are related to its codebooks $\boldsymbol{C}^a$ as,

$$
{}^k a^{K \cdot t+k} = [\; \frac{<\; {}^k c_n^a, f_a^{K \cdot t+k}\; >}{\|{}^k c_n^a\|_2 \; \|f_a^{K \cdot t+k}\|_2} \;], \;\; \text{where} \;\; {}^k c_n^a = \boldsymbol{C}^a[k, n, :], \;\; n = \{0, 1, \ldots, N\} \tag{2}
$$

The temporal steps $t$ are $\{0, 1, \ldots, T_v - 1\}$. Note that, for audio the $k^{th}$ codebook of $\boldsymbol{C}^a$ is linked with $k^{th}$ audio feature in $\boldsymbol{f}_a^t$. Eq 1 and 2 embed the audio and video information into their respective MSCs. A softmax across the number of codes gives the probability distribution of the relationship between the codebooks and the corresponding modality representations,

$$
{}^k p^t = [\; \frac{\exp{(\tau \cdot\; {}^k v_n^t)}}{\sum_{j=1}^N \exp{(\tau \cdot\; {}^k v_j^t)}} \;], \;\; {}^k q^{K \cdot t+k} = [\; \frac{\exp{(\tau \cdot\; {}^k a_n^{K \cdot t+k})}}{\sum_{j=1}^N \exp{(\tau \cdot\; {}^k a_j^{K \cdot t+k})}} \;], \;\; n = \{0, 1, \ldots, N\} \tag{3}
$$

$\tau$ is the temperature for the softmax function. These distributions are computed for each $k \in \{0, 1, \ldots, K - 1\}$. The modality-specific information captured by ${}^k p^t$ and ${}^k q^t$ are used to relate and associate the two modalities as well as retrieve the video representations using the audio representations.

**Audio To Video Representations** The bottom half in the right panel of Figure 1 shows the schematics for obtaining video representations using audio. The $k^{th}$ feature in $\boldsymbol{f}_a^t$ directly estimates "interleaved" representations for video using the $k^{th}$ codebook in $\boldsymbol{C}^v$,

$$
\hat{f}_v^{K \cdot t+k} = \text{linear} \left( \sum_{n=1}^N {}^k q_n^{K \cdot t+k} \cdot {}^k c_n^v; \; \theta_l \right) \tag{4}
$$

where $\theta_l$ are the parameters of the linear layer, in practice, this linear layer includes batch-normalization (Ioffe & Szegedy, 2015). The $\hat{f}_v^{K \cdot t+k}$ (instead) are concatenated with $f_a^{K \cdot t+k}$ and then decoded by the decoder to produce enhanced speech. Note that, in the base AVSE model $T_a$ video features are simply repeated to upsample by a factor of $K$ and then concatenated to audio features. TAME helps estimate video information at a lower temporal resolution, which can be crucial for precise replacement of video representations.

Clearly, the video encoder is discarded during inference and as long as the size and compute of the TAME module is significantly smaller than the video encoder, the whole model is much more efficient compared to the full audiovisual model.

### 3.3 TRAINING OBJECTIVES

**TAME Specific Losses** To train the proposed TAME module, we propose three different training objectives. First, we need to ensure that the relationship between the video features and video codebook $C^v$ is well-structured so that $C^v$ gets embedded with video information. This is achieved through self-modality recall of $f_v^t$ for each codebook in $C^v$, $^k\tilde{f}_v^t = \text{linear}(\sum_{n=1}^{N} {}^kp_n^t \cdot {}^kc_n^v; \theta_l)$. A reconstruction loss then guides the learning

$$\mathcal{L}_{v \to v} = \sum_{t=0}^{T_v-1} \sum_{k=0}^{K-1} \| {}^k\tilde{f}_v^t - f_v^t \|_2^2 \tag{5}$$

Next, a reconstruction loss between the estimated video representations $\hat{f}_v^{K \cdot t+k}$ and $f_v^t$ enforces retrieval of video information through audio representations.

$$\mathcal{L}_{a \to v} = \sum_{t=0}^{T_v-1} \sum_{k=0}^{K-1} \| \hat{f}_v^{K \cdot t+k} - f_v^t \|_2^2 \tag{6}$$

Lastly, we establish a cross-modal association by linking the two MSCs through the distribution captured by $^kp^t$ and $^kq^{K*t+k}$. Let $P^k$ (captured by $^kp^t$) and $Q^k$ (captured by $^kq^{K*t+k}$) be the distributions over the codes for $k^{th}$ codebook in $C^v$ and $C^a$ respectively.

$$\mathcal{L}_{C_a \to C_v} = \sum_{k=1}^{K} D_{KL}(P^k || Q^k). \tag{7}$$

The loss function in Eq 7, the distribution of codes in each codebook of $C^a$ matches the corresponding ones in $C^v$. This is necessary as the codebooks in $C^v$ are probed using audio representations embedded in $^kq^t$ to obtain video representations.

**Task Specific Loss Functions** The overall training of MUTUD includes task specific loss functions which in this case are speech enhancement losses. In this work, the output of the enhancement models are complex spectrograms ($E$) of the enhanced speech. The time-domain waveform ($e$) from $E$ is obtained using Inverse-Short Time Fourier Transform. The speech enhancement loss functions we use are

$$\mathcal{L}_{\text{task}} = \| E - C \|_1 - \text{SI-SDR}(e, c) \tag{8}$$

where $C$ is the complex STFT of target clean speech and $c$ is the time-domain target clean speech. The SI-SDR loss is defined as $\text{SI-SDR}(e, c) = 10 \log_{10} \frac{\|\alpha c\|^2}{\|\alpha c - e\|^2}$, where $\alpha = \frac{e^T c}{\|c\|^2}$. The enhancement losses in Eq 8 are computed using both $f_v^t$ and $\hat{f}_v^t$ as inputs to the decoder and the overall $L_{\text{task}}$ is sum of these losses. This is necessary to warrant that the video encoder learns meaningful representations in the end-to-end training.

The total loss function is

$$\mathcal{L}_{\text{MUTUD}} = \alpha \mathcal{L}_{v \to v} + \beta \mathcal{L}_{a \to v} + \gamma \mathcal{L}_{C_a \to C_v} + \lambda \mathcal{L}_{\text{task}}, \tag{9}$$

where $\alpha$, $\beta$, $\gamma$ and $\lambda$ are the weights given to each loss.

A few points are worth noting here. The TAME which is enabling MUTUD seamlessly fit into the base AVSE framework and can be easily adopted for many common multimodal methods and tasks. In our experiments, we evaluate MUTUD for 3 multimodal tasks; AVSE, audiovisual speech recognition (AVSR), and audiovisual active speaker detection (AV-ASD).

## 4 EXPERIMENTAL SETUP

In our experiments, we evaluate MUTUD under 3 multimodal tasks; AVSE, audiovisual speech recognition (AVSR), and ego-centric audiovisual active speaker detection (AV-ASD). AVSE is of key focus as this task is often desired to be deployed in real-time communication and on-device, which exacerbates the multimodal challenges outlined earlier in the paper.

**Datasets**: For AVSE and AVSR tasks, we utilize the LRS3-TED corpus (Afouras et al., 2018b), a large-scale audiovisual dataset for speech tasks. For AV-ASD task, we use *EasyCom*, a challenging real-world egocentric dataset (Donley et al., 2021). Overall, this allows for a comprehensive evaluation of MUTUD under a wide variety of acoustic and visual noise conditions. Please refer to the Appendix for further details on the datasets.

### 4.1 IMPLEMENTATION DETAILS FOR AVSE

**Data processing.** For LRS3, we crop the lip regions, resize the cropped frames into 88×88, and transform them to grayscale following Kim et al. (2021c). The audio, sampled at 16kHz, is converted into a spectrogram using a window size of 20 ms and a hop length of 10 ms. We augment the video data by applying random spatial erasure and time masking for effective modeling of the visual context (Mira et al., 2022).

All models are trained using noisy-clean speech pairs where, speech samples from LRS3 are mixed with noise samples from the DNS Challenge (Reddy et al., 2021) noise set. The noisy mixture is obtained by randomly mixing up to 5 different noise samples. The SNR range for mixing is -15 dB to 10 dB. We report results under 2 test conditions, (a) 3 background noises (3-BN) are present in the noisy mixture, and (b) 5 background noises (5-BN) are present. Evaluations are done at five different SNRs (in dB): 5, 0, -5, -10, and -15.

**Architectural and Training details.** The audio-only enhancement model is a U-Net architecture based on the gated convolutional recurrent network (GCRN) (Tan & Wang, 2019). The decoder includes an LSTM layer. The input to the model is a complex spectrogram of the audio. The Audiovisual model (Mira et al., 2023) is built on top of this audio-only model by employing a 3D convolutional layer followed by a ResNet18 (He et al., 2016) as the video encoder. The video and audio encoder outputs are concatenated and forwarded through the decoder to produce complex spectrograms of the enhanced audio. For MUTUD, $K = 4$ and we set the number of codes, $N$ in the MSCs to 32 after conducting an ablation study for different $N$ (Sec. 5.4). We train using AdamW optimizer (Kingma & Ba, 2014) with a learning rate of $10^{-4}$. We adopt a cosine scheduler (Loshchilov & Hutter, 2016), adding a warmup for 20 epochs. Loss function (Eq. 9) hyperparameters $\alpha$, $\beta$, $\gamma$ are simply set to 1.0 and $\lambda$ to 0.01. Please refer to the Appendix for more details.

**Evaluation metrics.** We utilize three standard speech quality and intelligibility metrics for AVSE: Short Time Objective Intelligibility (STOI) (Taal et al., 2010), Scale-Invariant Signal-to-Distortion Ratio (SISDR) (Le Roux et al., 2019), Perceptual Evaluation of Speech Quality (PESQ) (Rix et al., 2001).

## 5 RESULTS AND DISCUSSIONS

### 5.1 PERFORMANCE ANALYSIS

Table 1 presents quantitative results for the AVSE task under 3-background noise test condition. A few important details about the reported methods are in order. For a fair comparison, in addition to audio-only, we also report the performance of audio-only (*w. matched params*), that is, a model with number of parameters matched with MUTUD. This is important to establish that the proposed TAME module is in fact providing crucial information not present in the audio modality and cannot be compensated for by simply adding more parameters to the audio-only model. We show results for two versions of MUTUD representing two different training mechanisms: One where we train the entire model from scratch, denoted by *w.o. pretrained TAME*. Another, where we first pre-train the TAME module solely with clean audio and video frames and then fine-tune the entire model for the enhancement task. This is done to better guide the TAME module to store modality-specific information in the MSCs.

Table 1: Comparison of different models for 3-Background Noise test condition. Audio-only model is adopted from (Tan & Wang, 2019), and the corresponding Audiovisual model is adopted from (Tan & Wang, 2019) and (Mira et al., 2023)

| Method | STOI (%) | | | | | SISDR (dB) | | | | | PESQ | | | | |
|---|---|---|---|---|---|---|---|---|---|---|---|---|---|---|---|
| | 5 | 0 | -5 | -10 | -15 | 5 | 0 | -5 | -10 | -15 | 5 | 0 | -5 | -10 | -15 |
| Noisy Audio | 82.6 | 72.4 | 60.5 | 48.8 | 38.9 | 5.00 | 0.01 | -5.02 | -10.03 | -15.07 | 1.24 | 1.12 | 1.07 | 1.07 | 1.07 |
| Audio-only | 92.7 | 88.1 | 80.1 | 67.5 | 51.5 | 13.64 | 10.55 | 7.08 | 2.88 | -2.82 | 2.18 | 1.80 | 1.48 | 1.27 | 1.14 |
| Audio-only *w. matched params* | 93.0 | 88.3 | 80.4 | 68.0 | 52.6 | 13.75 | 10.58 | 7.05 | 2.86 | -2.62 | 2.30 | 1.87 | 1.53 | 1.30 | 1.16 |
| MUTUD *w.o. pretrained TAME* | 93.4 | 89.1 | 81.6 | 69.5 | 53.6 | 14.07 | 10.99 | 7.54 | 3.32 | -2.24 | 2.37 | 1.93 | 1.58 | 1.33 | 1.17 |
| MUTUD | 93.5 | 89.2 | 81.8 | 69.8 | 54.0 | 14.11 | 11.02 | 7.56 | 3.38 | -2.19 | 2.36 | 1.92 | 1.57 | 1.32 | 1.17 |
| Audiovisual | 93.5 | 89.6 | 83.3 | 74.0 | 62.7 | 13.92 | 10.90 | 7.61 | 3.81 | -0.86 | 2.35 | 1.94 | 1.60 | 1.38 | 1.20 |

Table 2: Number of Parameters and Multiply Accumulate Operations (MACs) for all models.

| | Audio-only | Audio-only *w. matched params* | Audiovisual | MUTUD |
|---|---|---|---|---|
| **# of Parameters** | 2.978M | 3.627M | 15.736M | 3.635M |
| **MACs** | 1.381G | 1.821G | 9.324G | 1.593G |

It is clear from Table 1 that our proposed framework MUTUD outperforms both the audio-only and the audio-only *with matched parameters* over all metrics and SNRs. This shows that the model has learned with visual information available during training and MUTUD is able to estimate video encodings and use them for better enhancement. It is worth mentioning that for extremely low SNRs of -10dB and -15dB, where multimodal models heavily rely on visual information for speech enhancement, MUTUD continues to consistently perform better than the audio-only model. This further highlights the TAME module's ability to estimate relevant visual information at prediction time. While we do not expect the MUTUD model to outperform or fully match the performance of the audiovisual model, it does an excellent job of reducing the gap between the unimodal and multimodal models. Except for extremely low SNR (-15dB), MUTUD is fairly competitive with the audiovisual model on all 3 metrics. This further argues for our multimodal training and unimodal deployment strategy. We also observe that the pre-trained TAME module is slightly superior to the one simply trained from scratch.

We observe similar trends for the more challenging 5-background noise test conditions showing that MUTUD can be successfully employed in such extreme noise conditions (shown in the Appendix). We also experiment with other audio-only, audiovisual, and the corresponding MUTUD models for AVSE. Please see the Appendix for additional results (Tables 6, 7, 8).

## 5.2 EFFICIENCY ANALYSIS

Table 2 shows parameter and Multiply Accumulate Operations (MAC) counts for all models. The MUTUD model is comparable in size and compute to both audio-only models. In fact, the MAC for MUTUD is around 13% lower compared to even the audio-only model with matching parameter count. However, we saw in Table 1 that MUTUD is much more superior compared to these models. With respect to the audiovisual model, MUTUD is smaller almost by a factor of 5 and has a smaller size and MAC by around 83% and 77% respectively. This shows the massive gain in efficiency one can achieve through our MUTUD learning framework.

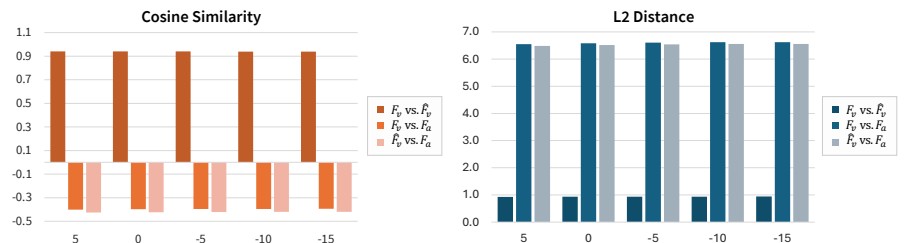

Figure 2: Cosine similarity (red) and $\ell_2$ distance (blue) between video features and estimated video features, video and audio features, and estimated video and audio features for different SNRs.

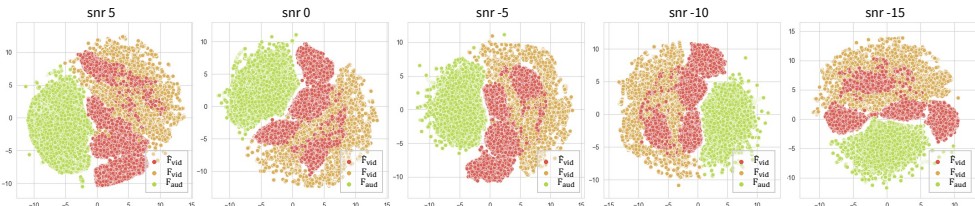

Figure 3: TSNE visualization of the estimated video features $\hat{F}_v$, the actual video features $F_v$, and the audio features $F_a$ for SNRs ranging from 5dB to -15dB.

### 5.3 TAME MODULE ANALYSIS

To analyze the estimated video features from the TAME module, we measure how similar they are to the original video and audio features. We compute the average cosine similarity and $\ell_2$ distance between video features and estimated video features ($F_v$ vs. $\hat{F}_v$), video features and audio features ($F_v$ vs. $F_a$), and estimated video features and audio features ($\hat{F}_v$ vs. $F_a$) for SNRs ranging from 5dB to -15dB. Figure 2 clearly indicates that the cosine similarity between the estimated video features and the original video features is high, around 0.94, while the similarity between audio and original (estimated) video features is low, $\approx -0.40$ ($\approx -0.42$). The $\ell_2$ distances showcase a similar trend where the audio and video features are further apart, and the estimated video features and the original ones are much closer. The high similarity between the estimated and original video features while having low similarity between the estimated video and audio feature evidence that TAME is not just a regurgitating audio feature but is actually functioning as designed (that is use audio information to get video information).

Furthermore, in Figure 3 we show the t-SNE visualization of the estimated video features, the original video features, and the audio features for all SNRs. Analyzing the clusters, we can clearly observe that the audio features $F_a$ form a distinct group, separate from the estimated video features $\hat{F}_v$ and the actual video feature $F_v$, demonstrating that the TAME can differentiate between modality-specific characteristics. More importantly, the estimated video features $\hat{F}_v$ and the actual video features $F_v$ are clustered closely together in the feature space, implying that the TAME module can accurately retrieve video features from the memory block, closely mirroring the actual video features even as the SNR levels decrease. Please see the Appendix for additional analysis of the TAME module.

### 5.4 ABLATION FOR CODEBOOK SIZE

We perform an ablation for the size of the codebooks in MSCs. We experiment with 4 different codebook sizes, $N$ (8, 16, 32, and 64) in each MSC of the TAME module. Table 3 indicates that as the size increases, more gain in speech enhancement performance is achieved, meaning that a larger number of codes in the MSCs can contain more meaningful features. We see that the $N = 64$ does not get much performance gain over $N = 32$. $N = 32$ is sufficient for embedding the audio and video information into the codebooks and then relating them to enable estimation of video representations using audio.

Table 3: Ablation on different numbers of codes, $N$, in each MSC of TAME.

| # of codes | STOI (%) | | | | | SISDR (dB) | | | | | PESQ | | | | |
|---|---|---|---|---|---|---|---|---|---|---|---|---|---|---|---|
| | 5 | 0 | -5 | -10 | -15 | 5 | 0 | -5 | -10 | -15 | 5 | 0 | -5 | -10 | -15 |
| 8 | 93.3 | 88.7 | 80.9 | 68.6 | 53.0 | 13.91 | 10.71 | 7.20 | 3.02 | -2.39 | 2.36 | 1.91 | 1.56 | 1.32 | 1.17 |
| 16 | 93.4 | 88.8 | 81.1 | 68.9 | 53.3 | 13.98 | 10.81 | 7.30 | 3.10 | -2.41 | 2.35 | 1.92 | 1.57 | 1.32 | 1.17 |
| 32 | 93.5 | 89.2 | 81.8 | 69.8 | 54.0 | 14.11 | 11.02 | 7.56 | 3.38 | -2.19 | 2.36 | 1.92 | 1.57 | 1.32 | 1.17 |
| 64 | 93.6 | 89.1 | 81.6 | 69.6 | 54.0 | 14.11 | 11.00 | 7.51 | 3.31 | -2.21 | 2.38 | 1.95 | 1.59 | 1.34 | 1.18 |

Table 4: Performance comparison on Audiovisual Speech Recognition (AVSR) task.

| Method | WER (%) ↓ | | | | |
|---|---|---|---|---|---|
| | 5 | 0 | -5 | -10 | -15 |
| Audio-only | 12.24 | 17.836 | 31.37 | 60.64 | 93.32 |
| MUTUD | 11.71 | 16.299 | 24.99 | 44.07 | 73.56 |
| Audiovisual | 5.26 | 7.088 | 11.01 | 21.12 | 36.56 |

Table 5: Performance comparison on Audiovisual Active Speaker Detection (AV-ASD) task.

| Method | mAP(%) |
|---|---|
| Video-only | 82.25 |
| MUTUD | 86.50 |
| Audiovisual | 87.60 |

## 5.5 AUDIOVISUAL SPEECH RECOGNITION (AVSR)

We additionally demonstrate the effectiveness of TAME through results on the AVSR task. For AVSR, we use V-CAFE (Hong et al., 2022) as the baseline architecture and measure speech recognition quality through Word Error Rates (WER). Please refer to the Appendix for full experimental setup details. Results are shown in Table 4. MUTUD while not outperforming the AV approach, shows a substantial reduction in WER compared to the audio-only method, highlighting TAME module's contribution in learning to leverage visuals even if it is available only during training. This is especially true for low-SNRs where visual play a more important role and MUTUD can help reduce WER by a considerable margin (6% for -5dB and 16% for -10dB)

## 5.6 AUDIOVISUAL ACTIVE SPEAKER DETECTION (AV-ASD)

To showcase our method's versatility, in the AV-ASD task, we assume the absence of audio modality at inference time, instead of the visual modality as done in previous experiments. We show results on the EasyCom dataset, a considerably more challenging real-world noisy dataset than LRS3. We design a network with 3D convolutional layers followed by an LSTM. Performance is evaluated using the mean Average Precision (mAP).

As seen in Table 5, the proposed model achieves a mAP score of 86.50%, which is a significant improvement over the Video-only method at 82.25% and quite close to the audiovisual approach at 87.60%. Notably, this outcome demonstrates the TAME capability to properly estimate audio representation using video representations, complementing the previously illustrated proficiency in video feature estimation. Thereby, reinforcing its applicability in different multimodal scenarios both in terms of tasks and datasets.

## 6 CONCLUSION

This work is motivated to address practical challenges in using multimodal solutions in real-world applications. We build and train the models keeping in mind that inference will be unimodal – a multimodal training but unimodal deployment strategy. In MUTUD, the model learns to associate and relate different modalities through modality-specific codebooks. Once this is achieved during training, the representations of modality absent during inference are obtained using the one present. Our framework and TAME are fairly generic and can be easily adapted for other common multimodal learning tasks and models. We can also extend MUTUD to more than two modalities through pairwise MSCs.

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

# A APPENDIX

## A.1 DATASETS

**LRS3**: For AVSE and AVSR tasks, we utilize the LRS3-TED corpus (Afouras et al., 2018b), a large-scale dataset of TED and TEDx videos. LRS3-TED consists of audio-visual pairs and corresponding text transcriptions for 151,819 utterances, totaling 439 hours. Following the original splits, we use ∼131,000 utterances for training and ∼1,300 utterances for testing. For AVSE, the clean speech samples are taken from LRS3 and the noise samples are taken from (Reddy et al., 2021) noise set. The videos are 25 fps with 224 × 224 resolution. During pre-processing, we center-crop at the mouth with a size of 88 × 88.

**EasyCom**: We employ EasyCom (Donley et al., 2021) for the AV-ASD task. This dataset contains ∼ 5 hours of natural conversations recorded in a noisy restaurant-like environment. The ego-centric nature of the data makes it extremely challenging as the sensory devices (camera and microphone on wearable glasses) are always moving. The ego-motions makes it difficult to learn from the video and the audio is corrupted by noise, making audiovisual active speaker detection (AV-ASD), challenging on this dataset. The dataset includes annotated voice activity, speech transcriptions, head bounding boxes, target of speech, and source identification labels. We use train-test splits from Hsu et al. (2022).

## A.2 ARCHITECTURAL AND TRAINING DETAILS

### A.2.1 AUDIOVISUAL SPEECH ENHANCEMENT (AVSE)

We mainly report the performance of the proposed Temporal Modality Retrieval module adopted from a Gated Convolutional Recurrent network (GCRN) (Tan & Wang, 2019). The input video frames are a shape of $T \times H \times W \times C$, where $T$ is the total frames of video, $H$, $W$, and $C$ are the height, width, and channel of a frame, respectively, and the input audio sequences are a shape of $F \times S$ where $F$ and $S$ represent frequency channels and frame length, respectively.

For the video encoder, we utilize a 3D convolutional layer followed by a ResNet18 (He et al., 2016). For the audio encoder, we adopted the GCRN, which is composed of two 2D convolutional layers, where the outputs of each convolutional layer, one followed by Sigmoid activation, are multiplied. Then, the concatenated audio features and video features are taken into a 2-layer Grouped LSTM. The decoder consists of 5 deconvolutional layers with a skip connection like a U-net architecture. The encoder-decoder structure is designed in a symmetric way, where the number of kernels progressively increases in the encoder and decreases in the decoder. To aggregate the context along the frequency direction, a stride of 2 is adopted along the frequency, dimension in all convolutional and deconvolutional layers. For another baseline shown in Table 7, we follow the same architecture and implementation details as Visualvoice (Gao & Grauman, 2021).

### A.2.2 AUDIOVISUAL SPEECH RECOGNITION (AVSR)

For the AVSR framework, we adopt V-CAFE (Hong et al., 2022) as a baseline architecture. The video encoder in the V-CAFE architecture consists of a 3D convolution layer with Batch Normalization and Max-pool followed by ResNet-18 (He et al., 2016), and the audio encoder contains two 2D convolution layers followed by one ResBlock for the audio front-end. The input shape of the model is the same as the Audiovisual Speech Enhancement model.

For the Visual Context-driven Audio Feature Enhancement module (V-CAFE), cross-modal attention followed by a noise reduction mask is applied. The noise reduction mask consists of two convolution layers with ReLU and Sigmoid activation respectively. The mask is multiplicated to the audio features $f_a$, and the masked audio features are summed with the original features to obtain the enhanced audio features. Finally, with the enhanced audio features and the visual features are concatenated with the linear layer and taken into

Conformer (Gulati et al., 2020) for the encoder and Transformer (Serdyuk et al., 2022) for the decoder for predicting the speech.

The Conformer (Gulati et al., 2020) sequence encoder is composed of hidden dimensions of 512, feed-forward dimensions of 2048, 12 layers, 8 attention heads, and a convolution kernel size of 31. The Transformer (Serdyuk et al., 2022) sequence decoder contains hidden dimensions of 512, feed-forward dimensions of 2048, 6 layers, and 8 attention heads are employed. Note that the video features are upsampled with the nearest neighbor interpolation to match the size of the audio features when taken into the proposed Temporal Modality Retrieval Module.

We follow the same implementation details reported in V-CAFE (Hong et al., 2022). We utilize background noises in diverse environments of DEMAND (Thiemann et al., 2013) dataset with SNR range randomly chosen from -15dB to 15dB for training. For testing, we report the testing performance at five different SNRs (in dB): 5, 0, -5, -10, -15.

### A.2.3 AUDIOVISUAL ACTIVE SPEAKER DETECTION (AV-ASD)

The AV-ASD model consists of a mouth keypoint detector to crop the lip region, CNN-based video and audio encoders, and a fusion layer followed by a causal temporal layer to incorporate a longer temporal past context. For the mouth keypoint detector, we adopt the ground truth facial per speaker manually checked by annotators. From the keypoints, we generate a new face crop by cropping the region by half of the width of the face region horizontally and also crop a quarter of the height downwards and three-quarters of the height upwards from the center of the mouth. We also generate a lip region, cropping the same way horizontally but cropping a quarter of height up and down from the mouth center.

The audio encoder is adopted from a VGG-M (Chatfield et al., 2014) operating on 13-dim Mel-Frequency Cepstral Coefficient (MFCC), treated as single-channel images. For the video encoder, we use a spatio-temporal VGG-M (Chatfield et al., 2014) composed of a 3D convolutional layer followed by a stack of 2D convolutions. We also adopt a Self-Attentive Pooling (SAP) layer like (Bhattacharya et al., 2017) for fusing the output audio and video features. Lastly, we set a unidirectional LSTM layer for temporal modeling to sequentially process consecutive embeddings from the fusion layers to predict speech activity corresponding to the latest frame followed by a projection layer and a sigmoid activation to derive activity predictions for each target speaker. When taken into the proposed Temporal Modality Retrieval Module, like the AVSR model, the video features are upsampled with the nearest neighbor interpolation to match the size of the audio features.

For the training, we apply horizontal flipping, random rotation within $-15° \sim +15°$, and motion blur augmentation with kernels randomly from 10, 25, 50, and 100. Due to the limited amount of dataset, we firstly pretrain the model with a larger dataset, VoxCeleb2 (Chung et al., 2018), to produce a better performance and generalization. We train using SGD optimizer (Robbins & Monro, 1951) with a learning rate of $5^{-5}$ with a weight decay of $5^{-4}$.

### A.3 ADDITIONAL RESULTS AND ANALYSIS

### A.3.1 ADDITIONAL RESULTS ON AUDIOVISUAL SPEECH ENHANCEMENT (AVSE)

Table 6 shows performance comparison for 5-Background noise test conditions. We see that the method shows trends comparable to the 3-noise case and consistently maintains better performance than audio-only in this more challenging noise scenario.

In Table 7 we show additional results after incorporating MUTUD within the VisualVoice (Gao & Grauman, 2021) framework. This demonstrates our method's flexibility with the underlying network architectures for the AVSE task.

Table 6: Performance comparison of different models for 5-Background noise test conditions.

| Method | STOI (%) | | | | | SISDR (dB) | | | | | PESQ | | | | |
|---|---|---|---|---|---|---|---|---|---|---|---|---|---|---|---|
| | 5 | 0 | -5 | -10 | -15 | 5 | 0 | -5 | -10 | -15 | 5 | 0 | -5 | -10 | -15 |
| Noisy Audio | 81.7 | 70.8 | 58.2 | 46.0 | 36.3 | 5.00 | 0.00 | -5.00 | -10.00 | -15.02 | 1.21 | 1.10 | 1.06 | 1.06 | 1.08 |
| Audio-only | 92.2 | 87.0 | 78.3 | 64.4 | 47.0 | 13.28 | 10.08 | 6.47 | 1.99 | -4.17 | 2.16 | 1.74 | 1.43 | 1.23 | 1.11 |
| MUTUD *w.o. pretrained TAME* | 92.7 | 87.7 | 79.3 | 65.5 | 48.1 | 13.45 | 10.32 | 6.72 | 2.27 | -3.88 | 2.24 | 1.8 | 1.47 | 1.25 | 1.12 |
| MUTUD | 92.8 | 88.0 | 79.6 | 65.6 | 48.0 | 13.60 | 10.43 | 6.85 | 2.33 | -3.86 | 2.23 | 1.80 | 1.47 | 1.25 | 1.12 |
| Audiovisual | 92.9 | 88.6 | 81.6 | 71.3 | 58.9 | 13.43 | 10.37 | 6.95 | 2.90 | -2.23 | 2.25 | 1.85 | 1.53 | 1.30 | 1.16 |

Table 7: Performance comparison with Visualvoice (Gao & Grauman, 2021) audio-only, audiovisual, and MUTUD for verifying TMR's robustness to other baseline architecture.

| Method | STOI (%) | | | | | SISDR (dB) | | | | | PESQ | | | | |
|---|---|---|---|---|---|---|---|---|---|---|---|---|---|---|---|
| | 5 | 0 | -5 | -10 | -15 | 5 | 0 | -5 | -10 | -15 | 5 | 0 | -5 | -10 | -15 |
| Noisy Audio | 82.6 | 72.4 | 60.5 | 48.8 | 38.9 | 5.00 | 0.00 | -5.00 | -10.03 | -15.06 | 1.24 | 1.12 | 1.07 | 1.07 | 1.07 |
| Audio-only | 91.8 | 85.9 | 76.5 | 64.0 | 48.8 | 11.67 | 8.62 | 5.17 | 1.08 | -4.90 | 2.05 | 1.60 | 1.32 | 1.17 | 1.09 |
| MUTUD | 93.5 | 89.0 | 82.0 | 71.3 | 56.5 | 12.72 | 9.68 | 6.53 | 2.98 | -2.04 | 2.40 | 1.94 | 1.58 | 1.33 | 1.18 |
| Audiovisual | 94.0 | 90.1 | 84.1 | 75.4 | 64.3 | 12.92 | 9.94 | 6.90 | 3.54 | -0.86 | 2.51 | 2.03 | 1.65 | 1.39 | 1.21 |

Table 8: Enhancement Performance comparison for models. The audio-only and the corresponding audiovisual used here are smaller and weaker.

| Method | STOI (%) | | | | | SISDR (dB) | | | | | PESQ | | | | |
|---|---|---|---|---|---|---|---|---|---|---|---|---|---|---|---|
| | 5 | 0 | -5 | -10 | -15 | 5 | 0 | -5 | -10 | -15 | 5 | 0 | -5 | -10 | -15 |
| Noisy Audio | 82.6 | 72.4 | 60.5 | 48.8 | 38.9 | 5.00 | 0.00 | -5.00 | -10.03 | -15.06 | 1.24 | 1.12 | 1.07 | 1.07 | 1.07 |
| Audio-only | 91.9 | 86.6 | 77.7 | 64.5 | 49.3 | 12.98 | 9.68 | 5.99 | 1.58 | -4.14 | 2.12 | 1.73 | 1.44 | 1.24 | 1.13 |
| MUTUD *w.o. pretrained TAME* | 92.3 | 87.3 | 78.9 | 66.0 | 50.3 | 13.31 | 10.07 | 6.45 | 2.08 | -3.65 | 2.20 | 1.79 | 1.49 | 1.27 | 1.14 |
| MUTUD | 92.3 | 87.3 | 78.9 | 66.0 | 50.5 | 13.29 | 10.04 | 6.43 | 2.06 | -3.66 | 2.22 | 1.81 | 1.49 | 1.27 | 1.14 |
| Audiovisual | 92.6 | 88.2 | 81.1 | 71.1 | 59.9 | 13.23 | 10.10 | 6.65 | 2.72 | -2.01 | 2.15 | 1.80 | 1.51 | 1.31 | 1.18 |

We analyze robustness of the TAME module with a smaller baseline architecture. To do so, we reduce the output dimension of the last two layers of the the audio encoder from 128 to 64. This results in audio-only and audiovisual models with 0.815M and 12.195M parameters, respectively. Results in Table 8 verify the robustness of TAME module which showcases quantitative trends similar to the original architecture.

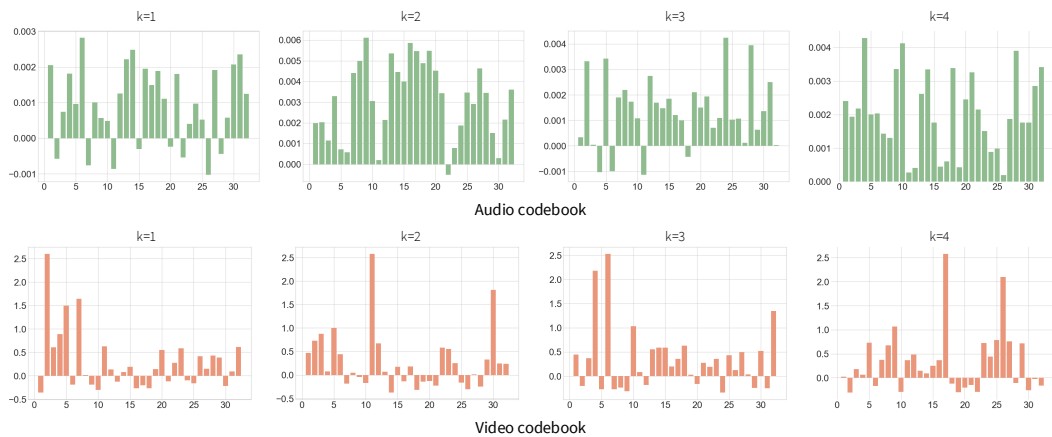

Figure 4: Data distribution of $k^{th}$ audio codebook and video codebook.

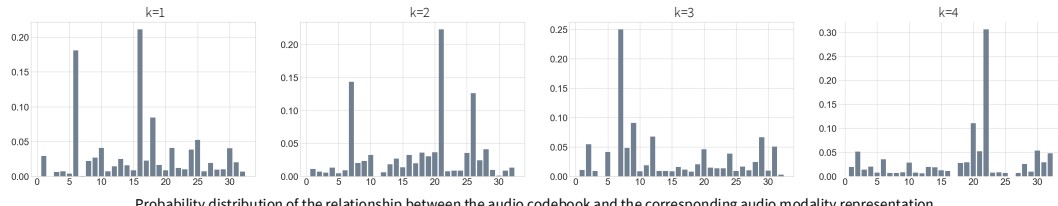

Figure 5: Probability distribution $q$ of the relationship between the $k^{th}$ audio codebook and the corresponding audio modality representation for estimating video representation for a sample noisy audio frame.

### A.3.2 ADDITIONAL ANALYSIS ON TAME MODULE

We analyze the distribution across the audio and video codebooks for all $K$. Figure 4 illustrates the distribution of data across the audio and video codebooks, with each $K^{th}$ code showing different usage patterns. This variation in usage across different codebook elements suggests that each code contains different audio and video information.

Additionally, we visualize the softmax probability distribution $q$ of the relationship between the $K^{th}$ audio codebook and the corresponding audio modality representation for a sample input test audio dataset, shown in Figure 5. The variation in these vectors further supports the conclusion that different parts of the codebook are being actively utilized.

### A.4 LIMITATIONS AND BROADER IMPACTS

While the MUTUD introduces a novel approach to multimodal training with the proposed TAME module and applies it to audiovisual speech tasks, its behaviour on other tasks and models in multimodal domain remains to be seen. So further exploration through even more complex tasks and models may shed more light on MUTUD. Regarding broader societal impacts, the MUTUD can be crucial for real-time, real-world applications (especially for on-device modeling), and not all kinds of information are available during real-world uses. This can make such applications more accessible as well as energy efficient.

