# OpenReview forum: "Efficient Audiovisual Speech Processing via MUTUD: Multimodal Training and Unimodal Deployment"
_ICLR.cc/2025/Conference — Submitted to ICLR 2025_

### Official Review · Reviewer_2ZCc · 2024-11-02

**Soundness:** 1
**Presentation:** 3
**Contribution:** 1
**Rating:** 3
**Confidence:** 3

**Summary:**

This paper tackles the interesting task of training on multimodal data and then evaluating using only one modality. This is achieved by predicting the other modality. Their method focuses on the use of code books to quantize audio and visual inputs, and they apply various loss functions in order to align the quantized modalities of the code books. They also align the dimensions of the audio and video via their interleaving method. They beat their benchmarks in several tasks while maintaining a small model footprint - measured by parameters and MACs.

**Strengths:**

The paper presents a useful approach to multimodal-to-unimodal distillation through its use of codebooks for modality alignment. The technical work includes benchmarking across different datasets and tasks, with testing under various noise conditions and SNR levels. The authors provide ablation studies on codebook sizes and architectural choices, and show how their method works with different baseline architectures. The experimental results are well-documented, and the paper's structure makes it easy to follow their methodology and findings.

From a practical standpoint, the work offers a solution for running multimodal systems with single modality inputs during deployment. Their implementation reduces model size by 80% and requires less computation compared to full multimodal systems. The authors' method for aligning audio and visual dimensions works effectively, and they demonstrate good performance across benchmarks while keeping the model compact in terms of parameters and MACs. The results suggest their approach could be useful for applications where resource efficiency matters.

**Weaknesses:**

There are several weaknesses to this paper. To summarise briefly: (1) the paper lacks significant contribution(s), (2) experiments do not support the authors’ claims. However, there are additional issues that I will discuss below. Firstly, point (1):

- A main motivation of this paper (motivation 2) is rather weak. A video camera available in even the cheapest smartphone is not “complex sensory devices working together seamlessly”. Other modalities, like text (unexplored in this paper) can be acquired via free open-source transcription models. There are many instances where multi-modal data is expensive and/or hard to obtain, but they do not tackle such use cases. I feel it would be better to focus the narrative on resource constraints in deployment settings (such as handling sensor failures).
They propose a framework that ideally maps from m training modalities to n inferences modalities, where n<m, but they focus on audio-visual training with only audio during inference time. It is unclear how MUTUD could generalise to other settings, when they do not even consider the reverse (only video at inference time). This is particularly true given their extensive usage of loss functions, which seem very task specific.
- The methods section is extremely short, and the majority of the work in this section is not the authors’ work. The largest section, about the codebooks, is simply defining what a codebook is. There is no additional contribution, and they do not cite the original work, -- it is unclear what the contribution of this section is. Similarly, the loss functions are not their work but are uncited. I believe the overall technique is unique, but a simple, direct application of others work is not a significant enough contribution to meet the requirements of ICLR.

Point (2):

- They compare to two methods: one adapted audio method from 2019, and one adapted audio-visual method from 2023. Experiments and resulting claims should be backed up by extensive experimentation that compares to many baselines. A 5 year old model that has been adapted is not sufficient to meet the requirements of ICLR.
- There is no empirical justification as to why estimating visual features from audio features is better than simply using only the audio features. In order to support this claim, their codebook methodology should be used in the audio only setting, the audio-visual setting, and the audio-visual setting with only audio at inference time. Comparing to other models means it's impossible to tell if the improvements are due to the audio-visual training, or simply because codebooks solve the task better.
- They only compare to adapted baselines. The tasks, such as AVSR and AVSE, are very common and well explored in the literature. They should compare to a list of contemporary methods, at least 6 contemporary audio-visual/audio only methods in total (i.e. 3 audio only, 3 audio visual).
- The number of parameters and MACs are interesting metrics, but have very little influence on inference speed. A large part of the paper’s narrative is discussing efficiency, yet none of the tables contain inference times. In modern deep learning, the biggest challenges are inference times and memory, reducing arbitrary values has little real-world utility.
- They do not break down specifically how the parameters and MACs are allocated, for example, how many MACs/Params for the audio code book, the video codebook, training size, evaluation size (on 1 modality).
- The tables are not well labelled, it is often unclear (without digging through paragraphs of experimental detail) what the evaluation task is for each table, on which dataset.

**Questions:**

You have audio to video but no video to audio, is this not included in the model?
From my understanding, codebooks are slow unless a transformer or similar architecture is applied in order to efficiently assign indexes in the codebook (i.e. https://arxiv.org/abs/2210.13438). Can you provide inference times in your tables?

Please additionally address the comments in the weaknesses section

---

> ### Author Response · Authors · 2024-12-01
>
> Thank you for your feedback. We address each of the main concerns and questions below.
>
> - **Video to Audio Experiments**:  We would like to highlight that we have provided video to audio experiments. The experiments for Active Speaker Detection are Video to Audio experiments, in which video modality is available during inference. Similar to the audio to video experiments, MUTUD is able to bridge the gap between the unimodal model and the audiovisual model. MUTUD gives an improvement of 5.2% over the video-only model and is almost 98.4% closer to the Audiovisual model in performance.
>
> - **Inference Times**:  First, we would also like to note that for certain applications (especially on-devices speech applications, e.g hearing aids, smart glasses etc.), parameter count is highly prohibitive in deploying multimodal models. We provide inference times for the models below. All inference times are measured for 10 seconds of audio. We will add inference times in the Table for compute. **As shown below the inference time of MUTUD is just 10% more than the audio-only model and almost 48% less than the audiovisual model. **
> -\
> | Method | Inference Time (ms)
> |:------------:|:---------: |
> Audio-only| 98.1 +/- 2.87|
> MUTUD| 108.0 +/- 2.1|
> Audiovisual| 206.0 +/- 8.1|
> - **Further Breakdown of Compute**: We are not sure what you mean by breakdown by “training size” and “evaluation size”. As for the breakdown for MUTUD w.r.t audio-only model - the only addition of compute by MUTUD is through the MSCs. The audio and the video codebooks are equally sized and hence their compute numbers are the same. The compute difference between MUTUD and the audio-only can be easily broken down. For example, the difference in the number of parameters between audio-only and MUTUD is 657K parameters (3.635M - 2.978M). We will explicitly put these numbers in the table to avoid any confusion.
>
> - **Experiments: Comparisons with other methods, adaptation of other backbones**:  **We would request you to please look at our detailed response for reviewer V27W**. We have provided detailed answers including comparison with other works, justification of uses of different models, independent evaluation on an out-of-domain audio-only dataset.
> If there are any remaining questions, please let us know.
>
> - **Method Section is Short**: We are not sure if we fully understand this question/concern. The TAME module (the architectural design that enables MUTUD) is core to this work and hence a large part of the method is devoted to it. The **TAME Specific loss functions**(Section 3.3, Eq 5, Eq 6 and Eq 7) to train TAME are also contributions of this work and are an equally important part of the method.
> We do use task-specific loss functions (e.g. SISDR for enhancement) which are well known in literature for that specific task. This is expected as we are trying to show the utility of MUTUD for different tasks and hence we do have to rely on task-specific losses. Task-specific loss functions ( are not central to this work and have not been emphasized. We did show them (e.g. Eq 8) for the sake of completeness. We will add a reference to the SISDR loss.
>
> - **Motivation of the paper**:
> – (1) Other multimodal tasks: The focus of this paper is indeed Audiovisual Speech Processing.  While MUTUD can be extended to other multimodal tasks (including texts), we focused on audiovisual tasks to keep the scope of this work well-defined. In our opinion, audiovisual learning itself is fairly broad and deserves to be explored thoroughly rather than focusing on all sorts of multimodal tasks. \
> (2) We would also like to emphasize that on-device use cases are not just limited to smartphones. There are plenty of devices (e.g hearing aids, headphones, smart glasses), where cameras are not available and compute could be severely limited, making it very difficult to obtain audiovisual data as well as deploy any form of audiovisual model. \
> (3) As mentioned above, we do consider “only video at inference” and one of our tasks (ASD) is based on that. \
> (4) We agree that sensory failures are one of the motivating factors. We would be happy to emphasize it further.

---

> > ### Author Response · Authors · 2024-12-01
> > **Official Comment by Authors (contd.)**
> >
> > - **Empirical Justification for Estimating Visual Features:**
> > (1) In environments where audio information is corrupted or incomplete, it is well-established in the literature that visual information can provide complementary cues to enhance speech understanding and related tasks, e.g [R1]. This principle underpins our rationale for the proposed model: leveraging visual information during multimodal training enables the model to learn richer representations that can be utilized for unimodal inference via TAME.\
> > (2) Moreover, we would also like to clarify that we are NOT comparing to other models. For example, the audio-only and the audiovisual models (say in Table 1) are closely related, the “audio” part of the audiovisual model is exactly the same as the audio-only model. Hence, it is fair to argue that  improvements by the audiovisual model is due to uses of visual signals. Similarly, the only difference between audiovisual and MUTUD is that MUTUD uses the TAME to estimate visual features (unlike  the audiovisual model where it is obtained from visual encoder). Hence, once again it is fair to compare MUTUD and audio-only. \
> > (3) The analysis of estimated video features (Figure 2 and Figure 3) as well as the codebook analysis (Figure 4 and Figure 5 in Appendix) further gives empirical justification of estimation and uses of visual features.
> >
> >
> > - **Table Captions:** We will make the table titles more explicit for clarity. We appreciate your feedback about it.
> >
> > [R1] Yang, Wenfeng, et al. "Research on robust audio-visual speech recognition algorithms." Mathematics 11.7 (2023): 1733.

---

### Official Review · Reviewer_V27W · 2024-11-03

**Soundness:** 2
**Presentation:** 3
**Contribution:** 2
**Rating:** 3
**Confidence:** 5

**Summary:**

The authors proposed a Multimodal Training and Unimodal Deployment (MUTUD) framework to train the model on audio and visual modalities, but only use audio modality for inferecnce. The authors expected the model to imagine visual information from extracted audio features to help downstream audio tasks, with a Temporally Aligned Modality feature Estimation (TAME) module. TAME comprises individual codebooks for audio and video, and is trained using reconstruction loss and KL divergence loss to associate the different modalities. During inference, the audio features are used to retrieve the corresponding video features through the codebooks. Other training objectives are used for specific downstream tasks.

**Strengths:**

The authors present a potential method to guide the projection from audio to visual representation when visual input is absent. However, the experiments are insufficient and lack persuasiveness.  The writing is clear and well-structured.

**Weaknesses:**

1. The author conducted experiments solely on the audio-visual dataset, comparing the performance of the multi-modal, audio-only, and MUTUD frameworks. However, no independent validation and comparison with other models were carried out on out-of-domain audio-only datasets. Such validation is crucial to checkout whether the model merely overfits to specific biases within the audio-visual dataset, or genuinely learns the projection from audio features to video features, thereby enhancing uni-modal tasks.

2. Lack of novelty. Similar methods have already been explored, such as "Multi-modality Associative Bridging through Memory: Speech Sound Recollected from Face Video," which predicts audio features from visual features.

3. The performance gain is quite limited compared to the audio-only model.

4. The experiment is not sufficient. Lack of comparison with some recent models on the three tasks.

5. Lack of validation for the effectiveness of the proposed TAME method on more advanced backbone.

**Questions:**

1. Could you provide the no independent validation and comparison with other uni-modal models on audio-only tasks? Such as speech enhancement, ASR, speaker detection.

2. The authors have built the models and baselines using relatively older models (e.g., GCRN, 2019). How does TAME perform when integrated with the latest backbone models?

3. What are the performance outcomes and discussions in comparison with the latest state-of-the-art works across the three tasks?

---

> ### Author Response · Authors · 2024-12-01
>
> Thank you for your feedback. Below, we address your concerns and questions. Most importantly, we provide the independent validation on an out-of-domain audio-only dataset as well as provide clarifications and details on the uses of backbone models and relation with state-of-the-art for the backbones.
>
>
> ### **Independent validation on an out-of-domain audio-only dataset**
> Thanks for raising this point. We agree that evaluations on an out-of-domain audio-only dataset wiould add value to the paper. Below we provide additional results for this. We will update the paper with these results. **Results show that MUTUD improves the audio-only approach on this out-of-domain audio-only dataset as well**. This shows that the results are not overfits to specific biases within the audio-visual dataset.
>
> We evaluated MUTUD and the audio-only models on the DNS Challenge dataset. DNS Challenge [R1] is one of the most well-known dataset for speech enhancement tasks and for benchmarking SE models. We conducted experiments on both the synthetic set and the real world recording sets of DNS Challenge.
>
> Results on DNS Synthetic Set (Without Reverb)
> | **Method**     | **PESQ** | **SI-SDR** | **STOI** |
> |:------------:  |:---------: | :---------:| :---------:|
> | Noisy | 1.58 | 9.07 | 0.92 |
> |Audio-only (Ours) | 2.32 | 16.24 | 0.94 |
> | MUTUD (Ours) | 2.56 | 17.50 | 0.96 |
>
> For the real-world recordings, ground truth clean speech signals are not available for the test samples. Hence, we report the DNSMOS metrics as done by other works using this test set.
>
> Results on the Real Recordings for DNS
>
> | **Method**   | **MOS (P808)** | **OVRL** | **SIG** | **BAK** |
> |--------------|----------------|----------|---------|---------|
> | Noisy        | 3.05           | 2.26     | 3.05    | 2.50    |
> | Audio-only   | 3.30           | 2.66     | 3.19    | 3.34    |
> | MUTUD        | 3.41           | 2.70     | 3.19    | 3.43    |
>
>
> [R1] The INTERSPEECH 2020 deep noise suppression challenge: Datasets, subjective testing framework, and challenge results, Interspeech.

---

> > ### Author Response · Authors · 2024-12-01
> > **Official Comment by Authors (contd.)**
> >
> > ### **Experiments: Backbone models, Comparisons with other works, Performance gains**
> > **Different Backbone Models and State-of-the-art**
> >
> > - First, we would like to highlight that the **key motivation for the paper is not to necessarily beat state-of-the-art for any of the speech tasks**. It is to demonstrate that MUTUD can bridge the performance gap between unimodal models and multimodal models while keeping the overall model much more computationally efficient. Our experiments are designed to effectively demonstrate that.
> >
> > -  **For AVSE, we have shown experiments using 3 different backbones (Table 1, Table 7 and Table 8)**. **Please see Appendix in the paper for additional results.** The backbones include those inspired from GCRN/AVGCRN as well as VisualVoice (Gao & Grauman, CVPR), together providing various degrees of performances. Overall, the results on these 3 backbones shows that MUTUD can be applied to different backbones and is an effective approach.
> >
> > - The GCRN inspired SE backbones we used in this work is **causal and relatively small**, making them suitable for computationally efficient and on-device uses, where our MUTUD approach might find more applicability. Moreover, our carefully designed GCRN inspired **audio-only backbone (and the corresponding AV model)  is strong and comparable to state-of-the-art models**. Below, we show a comparison of this backbone on the DNS Challenge [R1] dataset which is widely used for benchmarking speech enhancement systems. Our comparison includes state-of-the art models of similar sizes (w.r.t # of parameters) and 1 with an order of magnitude larger size.
> > Note that **some of these SOTA models are not causal and our backbone is not trained on DNS Challenge**, which severely disadvantages our model.  Still, the performance of our backbone model is comparable to these models. Our backbone model trained on DNS Challenge is on par with these other models.
> > -\
> > | **Method**     | **PESQ** | **SI-SDR** | **STOI** | params count | causal
> > |:------------:|:---------: | :---------:| :---------:|:---------:|:---------:|
> > | Noisy | 1.58 | 9.07 | 0.92 | - |  -
> > |Audio-only (Ours) | 2.32 | 16.24 | 0.94 | 2.98M | Y
> > | MUTUD (Ours) | 2.56 | 17.50 | 0.96 | 3.63 | Y
> > |FullSubnet [R2] | 2.77 | 17.29 | 0.96 | 5.6M | N
> > |CleanUNet-Lite [R3] | 2.19 | - | 0.96 | 2.19M | N
> > |Demucs [R4] | 2.66 | – | 0.97 | 33.53 M | Y
> > |NSNet [R5] | 2.15 | 15.61 | 0.94 | 5.1 M | Y
> >
> > - For AV-ASD our model is actually state-of-the-art. Below we show comparison with other recent  AV-ASD works on EasyCom dataset. Note that, for AV-ASD, the video modality is used for MUTUD.
> > -\| **Method**     | **mAP (%)** |
> > |:------------:  |:---------: |
> > | TalkNet [R5] | 79.9
> > | SPELL+ [R6] | 85.9
> > | VideoOnly (Ours) | 82.25
> > | AudioVisual (Ours) | 87.60
> > |MUTUD (Ours)| 86.5
> >
> > - For AVSR For AVSR, our backbone model, is close to the state of the art especially when we factor in that it is trained only on LRS3. We used this model given it’s simplicity and strong performance.
> > In our paper (Table 4), we reported the performance for noisy cases, to provide more insightful results. On the clean case, the WER for our model is 2.9%
> > -\
> > | **Method** | **Training Data (Hours)** | **WER (%)**
> > |:------------:  |:---------: |:------------: |
> > Se2Seq[R7] | 438 (only LRS3) | 2.3
> > RNN-T [R8] |  31000 | 4.8
> > AV-Hubert [R9] |  1759 | 1.3
> > ViT3D-CM [R10] |  90000 | 1.6
> > Ours |  438 (only LRS3) | 2.9
> >
> > ### **Summarizing Experimental Rigor**
> > We would like to clarify and emphasize the breadth and rigor of the experiments conducted in our work which involves 3 distinct tasks and two datasets, carefully chosen to cover a broad range of challenges and applications within audiovisual speech processing. It involves 3 distinct tasks (speech enhancement, speech recognition, and active speaker detection), all of which are well-established and critical tasks in audiovisual speech processing. It also involves 2 different datasets, including a unique real-world-dataset of EasyCom. In our experiments, we also covered a variety of acoustic and visual conditions. On the audio end, various noise conditions (3 and 5 background noises, various levels of noises (SNR conditions)) and visual variations naturally present in these datasets – including non-front head, occlusions, motion-blur, ego-motion, and so on.
> >
> > The EasyCom dataset is particularly noteworthy as it is a real-world, ego-centric dataset capturing conversations in challenging acoustic environments. This dataset includes considerable motion blur, ego-motion, and varying lighting conditions, which closely mimic the complexities encountered in real-world scenarios.
> >
> > Given the extensive variability within our selected tasks and datasets, we believe the current set of experiments provides strong evidence for our framework’s efficacy in a variety of tasks as well as visual and acoustic conditions. We hope this also clarifies the reviewer’s comment on the questions and limitations.

---

> > > ### Author Response · Authors · 2024-12-01
> > > **Official Comment by Authors (contd.)**
> > >
> > > ### Improvements over Audio-only model.
> > > Our model is actually providing considerable improvements over the audio-only and bring the performance considerably closer to the AV model (upper bound). Below we summarize the improvements in STOI with respect to where the performance stands with respect to the AV model's performance. We show what relative level of performance (percentage of AV’s performance, AV is at 100) audio-only and MUTUD  gives.
> > >
> > > As we can see, MUTUD can recover a significant level of performance compared to the audio-only model, for all SNRs. In most cases, it is giving a massive boost to over the audio-only model and MUTUD is competitive with AV with the benefit of computational efficiency.
> > >
> > > | **Method**     | **5dB** | **0dB** | **-5dB** | **-10dB** | **-15dB**
> > > |:------------:|:---------: | :---------:| :---------:|:---------:|:---------:|
> > > |Audio-only | 92.7 | 91.3 | 86.0 | 74.2 | 52.9
> > > |MUTUD | 100.0 | 97.7 | 93.4 | 83.3 | 63.4
> > > |Audiovisual] | 100.0 | 100.0 | 100.0 | 100.0 | 100.0

---

### Official Review · Reviewer_Cmc7 · 2024-11-03

**Soundness:** 2
**Presentation:** 3
**Contribution:** 2
**Rating:** 5
**Confidence:** 4

**Summary:**

This paper proposes an approach to leverage multiple modalities in model training to improve performance when only a single or a subset of modalities are present at test time.  This is achieved by estimating the missing modalities based on their correlation with the known modalities and using those estimates in the inference process.  A codebook based approach to model the modality correlation is presented.  Results of three audio-visual tasks — speech enhancement, speech recognition, and active speaker detection — demonstrate that by using both modalities at training time while only one at test time results in a significant gains over baseline single modality performance on each of the three tasks.

**Strengths:**

* A novel approach to utilize information from two or more modalities at model training time to improve run-time performance when only a single or a subset of modalities are present.  Proposed approach relies on estimating, in a time aligned manner, modality features that are not present at run-time.
* Proposed model is parameter and compute efficient
* Results in a significant accuracy gains without adding too many parameters over the baseline unimodal model.

**Weaknesses:**

* Missing baselines: Using multiple modalities at training time only to improve performance under a single or a subset of modalities has been studied before.  E.g. [1].  References & comparison with those methods needs to be carried out to fully assess the merits of the proposed model.

[1] Abavisani et al., “Improving the Performance of Unimodal Dynamic Hand-Gesture Recognition with Multimodal Training”

* Only the case of audio-visual ASR / Speech enhancement / Active speaker selection is studied.  This provides a limited, bi-modal view of general multimodal tasks.  A more general multi-modal setting will make this work much more attractive.

**Questions:**

* Typo on Line 058: defnitely
* Lines 195-196: define ‘D’

---

> ### Author Response · Authors · 2024-11-27
> **Response by Authors**
>
> **Questions** \
> Thank you for pointing out the typo and the missing definition. We will update the manuscript to correct these.
> \
>
>
> **Missing baselines**
>
> We acknowledge the relevance of [1] by Abavisani et al. (“Improving the Performance of Unimodal Dynamic Hand-Gesture Recognition with Multimodal Training”), and we will reference this work as related literature in our paper.
>
> However, the context of our work is different which leads to critical differences, making it unfair and impossible to compare with this work.
>
> - **Context and Multimodal Learning Setting, Generalizability of Applications**: 1] is designed as a form of knowledge transfer, where multimodal learning is primarily a decision level fusion approach.  In [1] two separate hand-gesture recognition models are trained; one using RGB images and the other using depth images.  An additional objective function facilitates knowledge transfer between these models, and only one of the models is subsequently tested. This severely limits systems to which this approach can be applied. \
> Our multimodal learning setting is not a decision level fusion system. We are not training individual models using each modality. In fact, such an approach may not apply at all for some audiovisual speech tasks such as speech enhancement. Hence, it is not possible to directly compare with [1].  \
> We are focusing on audiovisual speech tasks where the two modalities have complex dynamic inter-connections and state-of-the-art systems have a single model taking in both modalities and combining features of audio and video in various ways. MUTUD is designed to be a simple and efficient approach to achieve unimodal deployment and can be dropped into a large number of multimodal methods with ease. We show that our approach can be applied to a variety of audiovisual tasks, example speech recognition, speech enhancement and active speaker detection.
>
> - **Baselines in Paper**: We did cover various baselines by including different audiovisual frameworks for enhancement. This includes 2 different backbones of AVGCRN [Mira et. al, 2023] and VisualVoice (Gao & Grauman, 2021). Please see Appendix for these results.
>
> - **Knowledge Transfer**: Lastly, our attempt to do knowledge transfer from audiovisual speech enhancement model to audio-only speech enhancement model did not show any significant improvement for the audio-only model. This shows that for complex tasks such as speech enhancement, such approaches might not work.
>
> - **Generalizability due to Architectural Contribution**: Our proposed Temporally Aligned Modality Feature Estimation (TAME) module is novel in its ability to estimate missing modality features at inference, improving unimodal performance while maintaining computational efficiency. \
> We show the flexibility of MUTUD with baseline architectures by incorporating it within the VisualVoice (Gao & Grauman, 2021) framework for the audiovisual speech enhancement (AVSE) task. This integration demonstrates that MUTUD is not tied to a specific architecture and can generalize effectively across different network designs. Results from this experiment highlight comparable performance improvements, reinforcing the versatility of our approach.

---

> > ### Author Response · Authors · 2024-11-27
> > **Response by Authors (contd.)**
> >
> > **A limited, bi-modal view of general multimodal tasks.**
> >
> > We agree that it will be interesting to explore our framework beyond audiovisual speech tasks. However, to keep the scope of this work well-defined and provide comprehensive evaluations and insights, we focussed on tackling several important tasks within the area of audiovisual speech processing. Please note that within audiovisual speech tasks we have done comprehensive study over 3 different tasks.
> >
> > Moreover, we would like to highlight the flexibility inherent in TAME and its potential applicability beyond audiovisual tasks to other multimodal tasks. By design, TAME allows for a many-to-one or one-to-many correspondence between features of different modalities. In our current implementation, each video feature corresponds to multiple (four) audio features, demonstrating this flexibility. This design principle could theoretically extend to other types of multimodal data, allowing TAME to handle both synchronous or asynchronous inputs effectively. In cases involving text, image, or sensor data, the TAME module could align and map features across asynchronous or non-continuous modalities, as long as suitable feature representations are available. Text embeddings can also be mapped to the audio features and TAME can be adapted to learning from audio-text as well. The interaction/fusion between modalities could be anything – concatenation, cross-modal attention, etc. and TAME can still be adapted. As mentioned above, we have not empirically explored these additional modalities and interaction between them to keep the scope well-defined,  it remains a promising direction for future research.
> >
> > To address this concern in the manuscript, we will clarify the generality of the MUTUD framework in the methods section and discuss the potential for extensions to other multimodal configurations in the discussion and future work sections.

---

### Official Review · Reviewer_YNYM · 2024-11-04

**Soundness:** 2
**Presentation:** 1
**Contribution:** 2
**Rating:** 5
**Confidence:** 5

**Summary:**

This paper proposes MUTUD, which enables unimodal inference (deployment) by leveraging multi-modal training (specifically audio-visual training in this work). The TAME module is proposed to enable cross-modal transfer and is proven to be effective. It achieves better results than single-model training while using fewer parameters than multimodal training across three speech tasks: speech recognition, active speaker detection, and speech enhancement.

**Strengths:**

1. The presentation is quite clear, and the design of TAMU is simple and easy to understand.
2. While there are few works focused on efficient unimodal deployment, this appears to be the first adaptation in audio-visual speech tasks. The motivation is therefore innovative.
3. The experiments are sufficiently extensive.

**Weaknesses:**

Major Concerns:
While I understand that multi-modal approaches can improve performance, they also increase costs, which motivated MUTUD's development. I am curious about practical utility in industry applications - would we really use audio-visual data for performance boosting? I would love to see actual industry statistics. Though the motivation is quite clear, I would appreciate seeing the "motivation behind the motivation."
Audio and video data, though time-synchronized, aren't always perfectly aligned per frame. I'm not sure if you have any insights to handle this. Would time-accurate alignment in TAME lead to error accumulation?
Additionally, I'm uncertain about how general TAME is. What if we have different video modalities like face or lip?


Minor Concerns:
On line 36, the claim that "visual modality is the most widely used in these speech tasks" needs evidence. From my perspective, text appears to be more prevalent.
Regarding lines 45, 49, and 54, the three bullet points seem to apply specifically to audio-visual multi-modal scenarios, while speech-text multimodal situations might be different. One important missing challenge is that multimodal input doesn't always bring improvement - it can be noisy and confusing. For example, face and speech are not "perfectly" matching.
On line 137, the related work section should consider "ReVISE: Self-Supervised Speech Resynthesis with Visual Input for Universal and Generalized Speech Enhancement." The literature review needs to be more comprehensive.

**Questions:**

See Weaknesses

---

> ### Author Response · Authors · 2024-11-26
> **Response by Authors**
>
> We sincerely thank the reviewer for their comments and feedback. Below, we address each of the reviewer’s concerns:
>
> **Practical Utility in Industry Applications**
>
> Audio-visual models have shown significant impact in areas such as video conferencing, real-time transcription in noisy environments, and assistive technologies for the hearing-impaired (hearing aids). For example, lip-reading systems in accessibility tools and AI-driven video production pipelines often rely on visual input to enhance audio understanding under challenging conditions. Similarly, speech tasks in noisy conditions (enhancement, transcription etc.) on devices with cameras  can be significantly improved by using visual inputs along with the audio. [R1, R2, R3, R4, R5, R6] are a just few prior works emphasizing on various applications. A notable study by Schwartz et al. (2004) [R7] highlighted the importance of visual cues in noisy speech perception, which directly aligns with our motivation.
>
> Our “motivation behind motivation” from applications perspectives is to enable these applications on devices, where compute requirements can pose significant deployment challenges. Our goal is not to focus on showing why multimodal learning is useful, rather solve the practical problem of deployment of multimodal models to devices. The paper is also well grounded as their are theoretical arguments to support that multimodaly trained systems can do better on unimodal data compared to unimodaly trained models [R8]. We will add specific examples and references to the manuscript.
>
> \
> \
> **Frame Alignment and Error Accumulation in TAME**
>
> Thank you for raising this interesting question. Our design inherently accommodates such temporal mismatches by employing a many-to-one or one-to-many correspondence between audio and video features, reducing the dependency on exact frame alignment. The MSCs formalism allows this flexibility. As in the presented case, multiple audio features across the time dimensions are used to produce “inter-leaved” video features (Line 220-231) at a given video time frame, allowing video-feature to be recalled appropriately for the downstream task even if there is some degree of misalignment.
>
> To demonstrate robustness, our training includes diverse noise and motion conditions captured under real-world conditions using a smart glass ( EasyCom dataset with motion blur and ego-motion).
> Empirical results suggest that TAME performs reliably even under moderate misalignments. While extreme misalignment can theoretically lead to error accumulation, empirical verification of the same is outside the scope of current paper. We also believe that alignment correction mechanisms for TAME, such as dynamic time warping, as a potential direction for future work. We will clarify this limitation in the discussion section.
>
> \
> \
> **Generality of TAME Across Modalities**
>
> In our current work, experimental results do show the method applied to facial inputs (LRS3). The EasyCom dataset takes it even one step further where the visual inputs are from real-world natural conversations. EasyCom includes significant variability in visual input (e.g., non-frontal faces, motion blur), demonstrating TAME’s adaptability.  We agree that extending TAME to other video modalities, such as lip movements, is a valuable avenue for exploration and TAME is designed as a general framework to associate features across modalities. This modularity allows it to adapt to various types of visual input as long as meaningful representations can be defined. We will include this point in the manuscript and highlight its potential for broader multimodal scenarios in future studies.
>
> \
> \
> **Minor Concerns**
>
> We thank the reviewer for these valuable points. Regarding Line 36, we want to clarify that in speech recognition tasks, visual input uniquely complements audio in noisy environments, enhancing robustness when acoustic signals are heavily corrupted. For Lines 45, 49, and 54, we will clarify that the challenges listed are specific to audio-visual setups and note that other multimodal configurations, like audio-text, may have distinct challenges, including noisy or mismatched inputs. We will also expand the related work section to include "ReVISE" and additional literature to provide a more comprehensive review.
>
> \
> \
> \
> [R1] Method and system for enhancing a speech signal of a human speaker in a video using visual information, US Patent\
> [R2] Audio-visual hearing aid, US Patent\
> [R3] Multimodal machine learning: A survey and taxonomy, IEEE PAMI\
> [R4] Looking to listen at the cocktail party: A speaker-independent audio-visual model for speech separation, ACM Transactions on Graphics\
> [R5] An overview of deep-learning-based audio-visual speech enhancement and separation, IEEE TASLP\
> [R6] Audiovisual speech processing, Book \
> [R7] Seeing to hear better: evidence for early audio-visual interactions in speech identification \
> [R8] A theory of multimodal learning, Neurips

---

### Author Response · Authors · 2024-12-02
**Request to respond to the rebuttal**

Dear Reviewers,
Thanks for your feedback and comments. We have provided detailed responses to your questions and concerns. Please take a look and let us know if there are remaining questions/concerns.

---

### Meta-Review · Area_Chair_xGVU · 2024-12-20

**Metareview:**

The manuscript has received four negative evaluations (two ratings of 5 and two of 3). Although the author have responded to most of the issues highlighted by the reviewers, certain critical concerns remain unresolved: 1) the method lacks novelty as similar approaches have been previously published (Reviewer V27W), and there are issues with unfair comparisons (specifically, the addition of codebooks to methods for other modality settings; Reviewer 2ZCc). Furthermore, the authors are advised to demonstrate how the inclusion of the TAME component enhances performance across different AV multimodal architectures to validate its effectiveness broadly. Despite showing some promise, the AC concurs with the reviewers' decision to reject this version of the paper. The authors are encouraged to incorporate the additional experiments presented during the rebuttal stage in their future submission.

**Additional Comments On Reviewer Discussion:**

The reviewers have primarily expressed concerns regarding the experiments presented in the paper. While the paper includes a series of ablation studies, the main issues highlighted involve unfair comparisons and the absence of certain ablations and analyses. Additionally, some reviewers have noted a lack of novelty, pointing out that similar methods have already been introduced. Although the authors have submitted additional results and explanations to bolster their case, key issues, such as the unfair comparisons and the novelty of the method, remain unresolved.

---

### Decision · Program_Chairs · 2025-01-22

Reject